# The Impact of Linear Filter Preprocessing in the Interpretation of Permutation Entropy

**DOI:** 10.3390/e23070787

**Published:** 2021-06-22

**Authors:** Antonio Dávalos, Meryem Jabloun, Philippe Ravier, Olivier Buttelli

**Affiliations:** Laboratoire Pluridisciplinaire de Recherche en Ingénierie des Systèmes, Mécanique, Énergétique (PRISME), University of Orléans, 45100 Orléans, France; meryem.jabloun@univ-orleans.fr (M.J.); philippe.ravier@univ-orleans.fr (P.R.); olivier.buttelli@univ-orleans.fr (O.B.)

**Keywords:** permutation entropy, time series, linear filters, preprocessing

## Abstract

Permutation Entropy (PE) is a powerful tool for measuring the amount of information contained within a time series. However, this technique is rarely applied directly on raw signals. Instead, a preprocessing step, such as linear filtering, is applied in order to remove noise or to isolate specific frequency bands. In the current work, we aimed at outlining the effect of linear filter preprocessing in the final PE values. By means of the Wiener–Khinchin theorem, we theoretically characterize the linear filter’s intrinsic PE and separated its contribution from the signal’s ordinal information. We tested these results by means of simulated signals, subject to a variety of linear filters such as the moving average, Butterworth, and Chebyshev type I. The PE results from simulations closely resembled our predicted results for all tested filters, which validated our theoretical propositions. More importantly, when we applied linear filters to signals with inner correlations, we were able to theoretically decouple the signal-specific contribution from that induced by the linear filter. Therefore, by providing a proper framework of PE linear filter characterization, we improved the PE interpretation by identifying possible artifact information introduced by the preprocessing steps.

## 1. Introduction

Information entropy was first proposed by Shannon in his seminal paper “A Mathematical Theory of Communication” [1]. This measure can effectively assess the amount of “surprise” (new information) contained in any given instance of the result of a random variable with a known distribution function. Originally formulated in the context of telecommunications, researchers have extended the applications of this technique in a wide array of research fields, such as econometry [2], computer theory [3], set theory [4], and medicine [5].

Several different entropy techniques and variants have been proposed since then. Tsallis’ entropy [6] and Renyi’s entropy [7] offer an alternative formulation to Shannon’s entropy, by modifying the overall weight of the probability distribution. Other entropy techniques are characterized by the particular definition of the event set, such as approximate entropy [8] and sample entropy [9]. One noteworthy example is Permutation Entropy (PE) [10], which measures the distribution of the ordinal patterns instead of the cardinal values of the signal. This approach is robust to noise and computationally efficient. Moreover, it requires no prior knowledge of the signal’s internal structure to successfully measure its information content.

Entropy techniques in general, and PE in particular, are almost never used on the raw signal directly. More often, the signal is first treated by a filtering procedure. There are many reasons for this, most notably noise reduction or the study of the signal’s dynamics in a particular frequency band. Some PE techniques, in particular multiscale PE [11] and its variants [12], implicitly incorporate a filtering step within their algorithms. Nonetheless, the applied filters are not free of unwanted effects, since any prior modification to the time series implies a modification in the final PE measurement. This effect has not been previously characterized in the literature. Even linear filters, widely implemented in signal processing, have an effect on PE, which is not well understood. For this purpose, we need to accurately outline the impact of said preprocessing and separate its contribution to PE from the signal’s information content.

In this work, we outlined the theoretical effect of linear filters over arbitrarily distributed time series. In particular, we proposed the characterization of the linear filter’s intrinsic PE, which explicitly describes the filter’s contribution to the final PE value, regardless of the original raw signal. This allows the correct interpretation of the PE algorithm over an arbitrary time series, by correctly discriminating the filter’s effect from the signal’s inherent information content.

The remainder of article is presented as follows: Section 2 introduces all the necessary background knowledge, including the definition of PE and the particular theoretical characteristics of PE for Gaussian time series. In Section 3, we develop the mathematical foundations of the linear filter’s intrinsic PE.

In Section 4, we test our theoretical results by applying a variety of linear filters (lowpass, highpass, and passband) over different correlated and uncorrelated signal models. Finally, we summarize and discuss our findings in Section 5.

## 2. Theoretical Framework

In this section, we outline the background of the PE original method in the context of signal processing. In particular, we briefly discuss the PE definition as proposed by [10,13]. We also discuss the theoretical PE in the particular case of Gaussian time series with stationary increments, proposed by [14].

### 2.1. Permutation Entropy

Permutation Entropy (PE) [10], in the context of time series, is based on the probability distribution of the possible ordinal patterns found within the signal. For a given embedding dimension *d* and a finite signal $x={\left[x_{1},x_{2},\ldots ,x_{N}\right]}^{T}$ of length *N*, we can extract N−(d−1) segments of length *d*. Each ordinal pattern corresponds to one of the d! possible permutations of the ranks of each segment.

If we assume no particular underlying model or process in the observed signal, the probability of each pattern can only be measured by means of computing the cardinality of each pattern within the series. Therefore, we can obtain an estimation of the pattern probability distribution as follows,
(1)$${\hat{p}}_{i}=\frac{\#\{n|\;1\leq n\leq N-\left(d-1\right),\;\left(x_{n},x_{n+1}\ldots ,x_{n+(d-1)}\right)\;\mathrm{has}\;\mathrm{pattern}\;i\}}{N-(d-1)},$$
where # denotes the count of pattern i∈{1,…,d!} within the signal [15]. Typically, this definition includes a downsampling parameter τ. In the present work, without loss of generality, we assumed no downsampling (τ=1).

By applying Shannon’s entropy definition [1] to this distribution, the PE of the system is computed as,
(2)$$\hat{H}=\frac{-1}{\ln(d!)}\sum_{i=1}^{d!}{\hat{p}}_{i}\ln{\hat{p}}_{i},$$
where the term ln(d!) ensures the value of $\hat{H}$ is normalized.

PE is a widely implemented method given its implementation’s simplicity and computational speed [10]. Moreover, this ordinal method is invariant to nonlinear monotonous transformations [10]. Nonetheless, for a precise pattern probability estimation, the method requires a sufficiently large signal length N≫d!.

### 2.2. PE for Gaussian Time Series

For a Gaussian process with stationary increments, Bandt and Shiha [14] proposed closed expressions for the pattern probabilities for d=3 and d=4, which depend only on the signal’s autocorrelation function. The expression for the first pattern probability for d=3 (in ascending order, such that $x_{t}<x_{t+1}<x_{t+2}$) is,
(3)$$p_{1}=\frac{1}{\pi }\arcsin\left(\frac{1}{2}\sqrt{\frac{1-\rho (2)}{1-\rho (1)}}\right),$$
where ρ is the normalized autocorrelation function of the discrete time lags λ=1 and λ=2. The aforementioned Gaussian process satisfies the following symmetries,
(4)$$\begin{array}{cc} p_{1} & =p_{6}\\ p_{2} & =p_{3}=p_{4}=p_{5}=\frac{1-2p_{1}}{4}, \end{array}$$
which completely define the probabilities for the event set. The case of d=4 is more intricate. The explicit pattern probabilities for this embedded dimension were explicitly shown in [14] as a function of the signal’s autocorrelation function. For dimensions d≥5, there is no closed expression for the pattern probabilities.

Any linear combination of Gaussian random variables is itself Gaussian. If we apply any linear filter (finite or infinite) to this signal, the result will still satisfy these properties [14]. For any linear Gaussian process, we can obtain its autocorrelation function, from which we obtain the pattern distributions and, subsequently, the theoretical expected PE values.

In the next section, we will exploit this property to obtain the linear filter’s intrinsic PE and isolate its effect from the signal’s original PE.

## 3. Methods

In this section, we develop the theory behind the intrinsic PE of linear filters. First, we briefly discuss the filtering process by means of the convolution operator, both in the time and frequency domains. We then apply an arbitrary linear filter on Gaussian noise (wGn). By taking advantage of the frequency properties of wGn and the results from Section 2.2 [14], we obtain the theoretical effect of linear filters in the signal’s PE.

### 3.1. Filters, Power Spectrum, and Autocorrelation

For any given signal x(t), we apply a filter function g(t) by means of the convolution operation:(5)$$y\left(t\right)=\left(x*g\right)\left(t\right)=\int_{-\infty }^{\infty }x\left(\alpha \right)g\left(t-\alpha \right)\;d\alpha ,$$
where α is a real continuous variable for integration. If we apply the convolution theorem in the context of Fourier transforms [16], we can describe the filtered signal in the frequency domain as follows,
(6)$$Y\left(f\right)=F\left\{(x*g)(t)\right\}=F\left\{x\left(t\right)\right\}F\left\{g\left(t\right)\right\}=X\left(f\right)G\left(f\right)$$
where the Fourier transform is defined as:(7)$$F\left\{x(t)\right\}=X\left(f\right)=\int_{-\infty }^{\infty }x\left(t\right)e^{-2\pi ift}\;dt.$$
and *f* is the frequency. In order to obtain the power spectrum of the signal x(t), it is enough to compute ${\left|X\left(f\right)\right|}^{2}$, the square of the module of X(f). Therefore, the filtered signal (x∗g)(t) will have a power spectrum of ${\left|X\left(f\right)G\left(f\right)\right|}^{2}$.

We were interested in obtaining the normalized autocorrelation function of (x∗g)(t). By means of the Wiener–Khinchin theorem [17], we know the inverse Fourier transform of the power spectrum is indeed the autocorrelation function of the filtered signal,
(8)$$\rho_{y}\left(\tau \right)=kF^{-1}\left\{{\left|X\left(f\right)G\left(f\right)\right|}^{2}\right\}$$
where τ is the continuous time lag and *k* is a constant with a chosen value such that $\rho_{y}\left(0\right)=1$. With the result of this theorem, we have a direct relationship between the Fourier transform of x(t) and the autocorrelation function $\rho_{y}\left(\tau \right)$. If x(t) is a Gaussian process, we can obtain the theoretical PE value of the filtered signal y(t) using Equations (Equation 2)–(Equation 4), for d=3.

### 3.2. Linear Filter’s Intrinsic Permutation Entropy

For the case of random signals, the power spectrum becomes a power spectral density by applying the expectation operator E[·]. In the particular case of the wGn, the power spectral density is flat and constant along all possible frequencies [16]. If x(t) is a wGn process, with variance $\sigma^{2}$ = 1, without loss of generality, the filtered signal y(t)=(x∗g)(t) power spectrum is,
(9)$$E\left[{\left|Y\left(f\right)\right|}^{2}\right]=E\left[{\left|X\left(f\right)G\left(f\right)\right|}^{2}\right]=c{\left|G\left(f\right)\right|}^{2},$$
since the filter is a deterministic component. If we compute the filtered signal’s autocorrelation function using Equation (Equation 8),
(10)$$\rho_{y}\left(\tau \right)=\rho_{g}\left(\tau \right)=kF^{-1}\left\{{\left|G\left(f\right)\right|}^{2}\right\}$$
where *k* is a real number such that $\rho_{g}\left(0\right)=1$.

The particular form of Equation (Equation 10) is revealing. By assuming uncorrelated white Gaussian noise, $\rho_{g}\left(\tau \right)$ is exclusively the autocorrelation function of the filter. If we insert $\rho_{g}\left(\tau \right)$ into Equation (Equation 3), we obtain,
(11)$$p_{1}=\frac{1}{\pi }\arcsin\left(\frac{1}{2}\sqrt{\frac{1-\rho_{g}\left(2T\right)}{1-\rho_{g}\left(T\right)}}\right),$$
where *T* is the sampling period of the signal. By subsequently computing the remaining pattern probabilities using Equation (Equation 4), we obtain a theoretical PE (d=3) value corresponding solely to the linear filter g(t).

Even when the calculations initially required the use of the wGn, the autocorrelation function $\rho_{g}\left(\tau \right)$ and the resulting PE value are not functions of *x*(*t*). Therefore, any linear filter has a corresponding intrinsic permutation entropy value. Moreover, this PE value can be obtained analytically, as long as the inverse Fourier transform of ${\left|G\left(f\right)\right|}^{2}$ has a closed form.

Since Equation (Equation 10) makes no reference to the distribution of x(t), any uncorrelated signal will lead to the same result, even if it is not Gaussian. This implies that the filter’s intrinsic PE corresponds to the maximum permutation entropy possible under the restriction of said filter. Therefore, any further reduction from this value must originate from the signal itself. If the effect of the filter is not taken into account, the final PE value can be mistakenly attributed solely to the signal’s dynamics.

### 3.3. Academic Filter Example

In order to better illustrate the computation of an intrinsic filter’s PE, we used the filter model provided by Farina and Merletti [18]. In particular, for a given wGn time series, the authors applied a bandpass filter with the following power spectrum,
(12)$$P\left(f\right)=\frac{kf_{h}^{4}f^{2}}{\left(f^{2}+f_{l}^{2}\right){\left(f^{2}+f_{h}^{2}\right)}^{2},}$$
with *f* being the frequency variable, *k* a real constant, and $f_{l}$ and $f_{h}$ the low- and high-cutoff frequencies, respectively.

The information provided by Equation (Equation 12) is enough to provide a theoretical PE for the aforementioned filter effect. First, we obtained the normalized autocorrelation function by means of the inverse Fourier transform of (Equation 12),
(13)$$\rho_{g}\left(\tau \right)=F^{-1}\left\{P(f)\right\}=\frac{2f_{h}}{{\left(f_{h}-f_{l}\right)}^{2}}\left(-f_{l}e^{-2\pi f_{l}\tau }+\frac{f_{h}^{2}+f_{l}^{2}}{2f_{h}}e^{-2\pi f_{h}\tau }+\pi \left(f_{h}^{2}-f_{l}^{2}\right)\tau e^{-2\pi f_{h}\tau }\right),$$
choosing again the constant *k* in such a way that $\rho_{g}\left(0\right)=1$.

In order to work with a discrete form of Equation (Equation 13), we needed to define the sampling period $T=1/f_{s}$, the inverse of the sampling rate. We subsequently used the autocorrelation function (Equation 11) to obtain the pattern probability distribution using Equations (Equation 3) and (Equation 4). Finally, we obtained the corresponding PE for d=3 using Equation (Equation 2).

The results shown in Figure 1b correspond to the theoretical predictions using Equations (Equation 2) and (Equation 13). We observed that the lowest PE curves corresponded to filters with narrow bandwidths, since few frequency components imply a signal with low complexity.

In order to compare these results, we generated 1000 uncorrelated white Gaussian noise signals of length $N=10^{5}$. We applied the filter in (Equation 12) on each series, using the parameters shown in Figure 1a. We subsequently measured their PE using Equations (Equation 1) and (Equation 2) directly. The results closely followed the curves in Figure 1b, within a 95% confidence interval. Note that, for shorter time series, we expected the experimental result to be slightly lower, since the PE estimator has a downward bias due to a finite number of data points [19].

## 4. Results and Discussion

In this section, we test the results obtained from Section 3, by means of time series simulations for different models, including white Gaussian noise, white uniform noise, and two Autoregressive and Moving Average (ARMA) models. Each signal was subject to a variety of lowpass, highpass, and bandpass linear filters. The resulting filtered signals were subject to the classic PE procedure (Equation 2) and subsequently compared to the theoretical predictions.

### 4.1. Filters on Simulated White Noise

In order to test the precision of the intrinsic filter’s PE (Equation 11), we performed a series of test on simulated signals. We created 1000 uncorrelated white Gaussian noise series of length $N=10^{5}$ each, sampled at a sampling frequency of $f_{s}=1000$ Hz. For each signal instance, we apply the filters specified in Table 1.

In Figure 2, we compare the theoretical PE (Equation 2), as a function to the cutoff frequency $f_{c}$, with the PE value obtained from simulations, using the moving average (which is part of the multiscale PE algorithm [20]), Butterworth, and Chebyshev type I lowpass filters from Table 1. Figure 3 repeats this experiment for highpass filters. Figure 4 shows a Butterworth (n=6) bandpass filter as a function of the low and high cutoff frequencies $f_{l}$ and $f_{h}$ respectively.

All proposed filters behaved almost identically with respect to their cutoff frequency, with the exception of the moving average filter in Figure 2b, which presented discontinuities. This was a consequence of the definition of the moving average filter, where the cutoff frequency $f_{c}$ was not an explicit parameter. Instead, $f_{c}$ was a discontinuous function dependent on the window length *L*, which was always a positive integer. In the case of the passband filters in Figure 4, all curves behaved similarly, regardless of the filter used (hence, only the Butterworth (n=6) is shown).

For all types of filters, the theoretical PE (Equation 2) curves lied between the confidence interval obtained from the simulations, at 95% confidence. As a general trend, all curves presented lower levels of PE when we reduced the passband bandwidth. This is straightforward to see for lowpass and highpass filters. Nonetheless, the bandpass filters’ PE curve is more difficult to interpret. Lower $f_{h}$ values led to lower PE values, but variations of $f_{l}$ did not necessarily lead to monotonic changes in PE. This implies both the passband frequency range and the actual frequency values play an important role in the filter’s information content.

We had similar results with other types of uncorrelated noise, even if the Gaussian assumption by Bandt and Shiha [10] did not hold. For example, we repeated the previous experiment with white uniform noise. As we can see in Figure 5, the behavior of PE (Equation 2) with respect to $f_{c}$ was identical to the case with white noise. This supported our claim from Section 3.2, where the particular distribution of the signal was not relevant in the computation of the intrinsic filter’s PE, as long as the signal was uncorrelated.

### 4.2. Filters on Correlated Gaussian Signals

In this section, we compute the PE of filtered correlated signals, in order to assess the contribution of the filter compared to the signal’s original PE. We used correlated Gaussian signals as a benchmark, since we could explicitly obtain theoretical PE values using Equations (Equation 4) and (Equation 11) for dimension d=3.

We proposed here the use of Autoregressive and Moving Average (ARMA) processes, since the resulting PE (d=3) has a closed expression, which is dependent only on the models’ parameters [21]. Moreover, ARMA processes are widely used to model complex phenomena, such as heart rate variability [22,23]. We simulate 1000 signals of length $N=10^{5}$ for each of the following models:Moving average process of order q=2, with $\theta_{1}=0.6$ and $\theta_{2}=0.4$ with the autocorrelation function:-$\rho_{MA}\left(1\right)=\frac{\theta_{1}+\theta_{1}\theta_{2}}{1+\theta_{1}^{2}+\theta_{2}^{2}}$,      $\rho_{MA}\left(2\right)=\frac{\theta_{2}}{1+\theta_{1}^{2}+\theta_{2}^{2}}$;Autoregressive process of order p=2, with $\phi_{1}=0.8$ and $\phi_{2}=0.15$ with the autocorrelation function:-$\rho_{AR}\left(1\right)=\frac{\phi_{1}}{1-\phi_{2}}$,      $\rho_{AR}\left(2\right)=\frac{\phi_{2}\left(\phi_{1}^{2}+1-\phi_{2}\right)}{1-\phi_{2}}$.

Since these processes comply with the Gaussian conditions in [14], we knew precisely the theoretical PE (Equation 2) we should expect for both models. The results are shown in Figure 6 and Figure 7. Figure 8 shows the PE difference between each filtered model’s signal and the maximum PE allowed by each filter, as a function of the cutoff frequency.

In Figure 6 and Figure 7, we observe, for all cases, that the PE of each model did not exceed the maximum theoretical curve of the intrinsic filter’s PE. This agreed with our theoretical results in Section 3. Furthermore, for both models, PE measurements obtained from simulated series agreed (with 95% confidence interval) with the theoretical expectations, where the PE values were obtained by computing the autocorrelation function of the filtered signal in Equation (Equation 8) and calculating the pattern probability distribution using Equations (Equation 3) and (Equation 4) for dimension d=3.

Figure 8 shows the explicit PE difference by applying the lowpass filters in Table 1 to the MA(2) and AR(2) models (Figure 8a,b, respectively). For these particular configurations, we clearly observe cutoff frequencies where the PE difference was the maximum. By applying lowpass filters with a normalized cutoff frequency of $f_{c}/f_{N}\approx 0.7$ (where $f_{N}$ is the Nyquist frequency) to the MA(2) model, we were able to identify the bandwidth that contained the most useful information. The same was true for the AR(2) model, where the normalized cutoff frequency was $f_{c}/f_{N}\approx 0.4$.

### 4.3. Discussion

When we empirically apply the PE algorithm to an arbitrary signal, we know that the pattern probability distribution is related to the signal’s autocorrelation function. If the signal is Gaussian, we can describe this distribution with a closed expression for dimensions d=3 and d=4, using the techniques already discussed [10]. The relationship is explicit only when the Gaussianity condition is satisfied. Nonetheless, autocorrelation and pattern distribution are still closely related for non-Gaussian signals, albeit not with a closed form. In the absence of a filter, the inherent dynamics captured by the PE correspond exclusively to the intrinsic characteristics of the signal.

The same cannot be said when we apply any type of preprocessing to the signal. By applying a filter we modify the autocorrelation function and, thus, the final PE result. The current literature acknowledges this effect, but does not offer an explicit estimation of the filter’s contribution to the final PE of the signal.

By obtaining the intrinsic maximum PE of any defined linear filter using Equation (Equation 10), we can confidently identify how much said filter reduces the PE measurements. Since this reduction describes the entropy of filtered uncorrelated white noise (Gaussian or otherwise), any further reduction from our measured PE must come from the signal itself. In fact, this is the maximum PE under the constraint of our defined filter.

The results from Figure 8 require further discussion. the presence of a clear maximum PE difference for specific cutoff frequencies implies the existence of specific bandwidths where most of the ordinal information content can be found. This, of course, depends on the specific model and characteristics of the signal, as well as the proper choice of the filter. This provides a useful reference when we need to identify the most relevant frequency regions for further analysis.

In practice, the signal’s internal dynamics are not the only source of PE deviation from the intrinsic filter’s PE. We still need to identify the contribution of the PE bias [19] to the final PE measurement. As long as we have a sufficiently long signal, and a reasonably low embedding dimension, we expect this bias to be low, or even negligible, compared to the contribution from the new dynamics of the filtered signal.

## 5. Conclusions

In this article, we outlined and characterized the effect of linear filter preprocessing on the time series’ Permutation Entropy (PE). First, by means of the Wiener–Khinchin theorem [17] and the ordinal pattern symmetries of Gaussian processes [14], we performed the theoretical development of the general linear filter’s intrinsic PE for low embedding dimensions. Next, to test the theoretical results, we applied a series of lowpass, highpass, and bandpass linear filters on the uncorrelated signals, such as white Gaussian noise and white uniform noise. Finally, we applied said filters to two Autoregressive and Moving Average (ARMA) processes of order two, in order to test the resulting PE for signals with intrinsic dynamics.

When plotting PE as a function of the filter’s cutoff frequency, we found our theoretical results to match the simulation results for all cases, with a 95% confidence interval. For the uncorrelated noise, we found no difference in the resulting curve, regardless of the signal’s distribution (Gaussian or uniform stochastic processes) for any given filter. This supported the theoretical result, where we established that the filter’s intrinsic PE curve was independent of the uncorrelated signal distribution. In the case of filtered ARMA processes, we observed lower PE values than those from filtered uncorrelated series. Moreover, the difference between both PE values were dependent on the cutoff frequency. When plotting said entropy difference, we found clear maxima for both ARMA models, with small variations for the different filters used.

The autoregressive and moving average filtered signals merit further attention. We can interpret the PE curve as the maximum entropy allowed by the filter, only achieved when we applied said filter to uncorrelated noise. Any further downward deviation from this curve corresponded to information contained within the signal. Moreover, this PE difference was dependent on the cutoff frequency. In the particular cases of the ARMA models used in the present work, we observed clear maximum PE values, which suggested an optimal passband that contained most of the ordinal information of the system. This may not be the case for an arbitrary signal, but the maximum PE difference, when present, led naturally to the proper choice of passband in order to extract the maximum amount of ordinal information.

With the results presented in this article, we are confident we can properly interpret the results from a filtered signal PE. If using a linear filter, we can separate the different sources of predictability: part of it comes from the filter preprocessing, and the other part comes from the underlying information contained in the signal. This is true in the case of general preprocessing filtering, as well as in cases where the filtering step is part of the PE algorithm, such as the multiscale PE variants [12,20]. It is important to note that, in practice, the PE is a biased statistic, which acts as an additional source of information, albeit not desired. For sufficiently long signals, this bias is negligible, but in the worst-case scenario, we should also take this effect into account for a proper interpretation of the phenomenon. There are still several challenges in the understanding of the filters’ effects on ordinal information, in particular the characterization of forbidden patterns [24] of filtered signals. This is a topic for future research to explore.

## Figures and Tables

**Figure 1 entropy-23-00787-f001:**
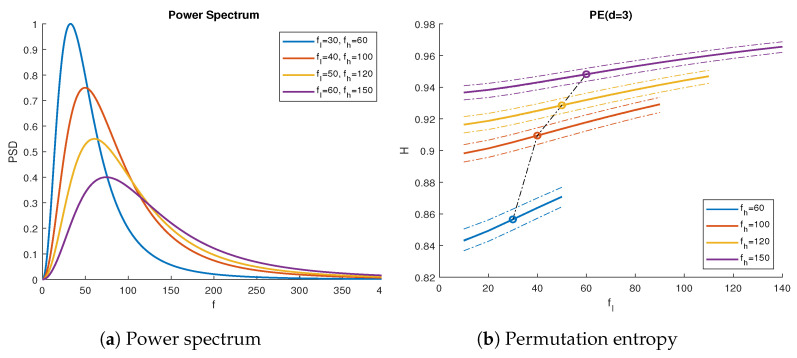
The wGn subject to the filter defined in (Equation 12). (**a**) Power spectrum of the filtered wGn with sampling frequency $f_{s}=1000$ Hz, for different values of cutoff frequencies $f_{l}$ and $f_{h}$. (**b**) Average permutation entropy for d=3 vs. $f_{l}$ for 1000 signal iterations of length $N=10^{5}$, showing curves for $f_{h}$. Solid lines represent theoretical results, while the dotted lines show the 95% confidence interval PE range from the simulations. The marked points represent the PE values corresponding to the power spectra in (**a**).

**Figure 2 entropy-23-00787-f002:**
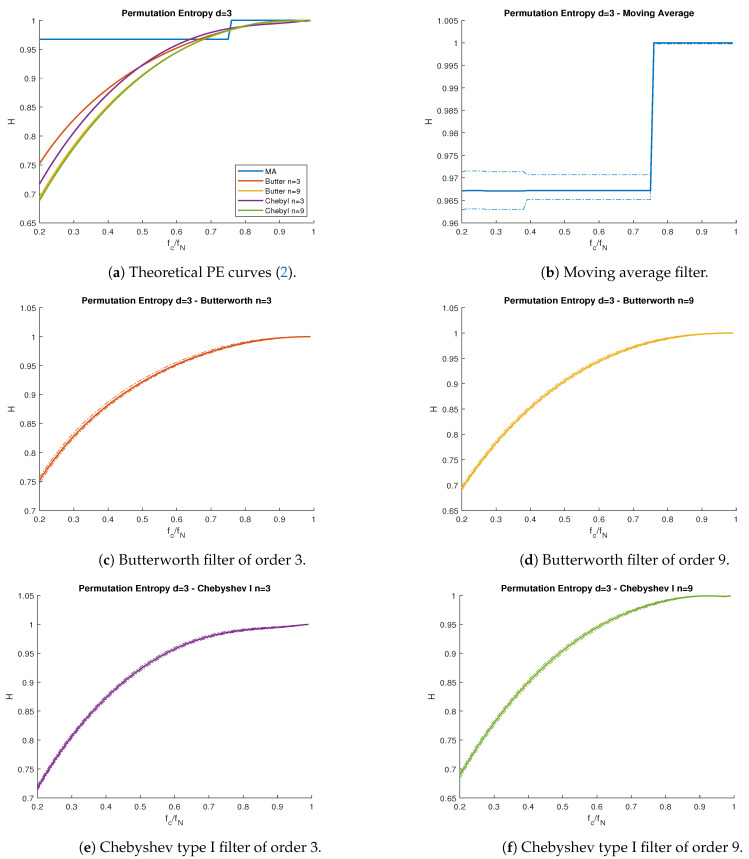
PE vs. normalized cutoff frequency $f_{c}/f_{N}$ for the wGn subject to lowpass filters. (**a**) Theoretical curves. (**b**) Moving average filter. (**c**) Butterworth filter of order 3. (**d**) Butterworth filter of order 9. (**e**) Chebyshev type I filter of order 3. (**f**) Chebyshev type I filter of order 9. Solid lines represent theoretical curves, while the band between dotted lines represent the simulations’ confidence interval for α=0.95. Nyquist frequency $f_{N}=500$ Hz.

**Figure 3 entropy-23-00787-f003:**
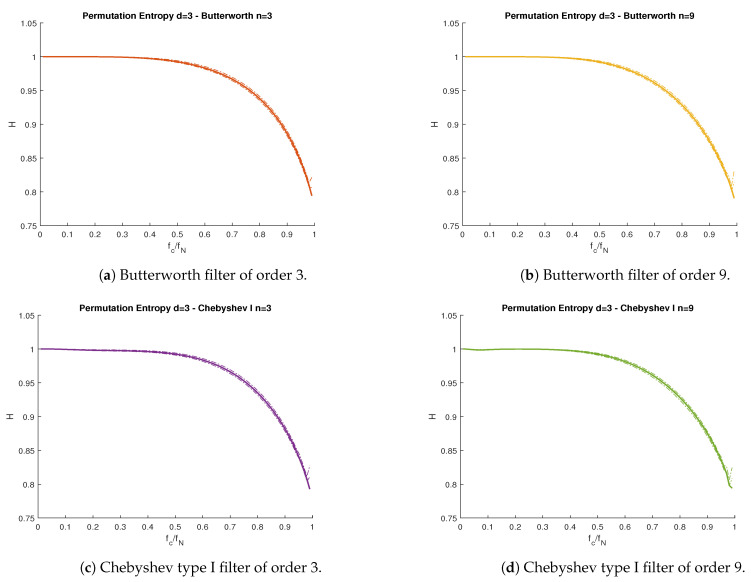
PE vs. normalized cutoff frequency $f_{c}/f_{N}$ for the wGn subject to highpass filters. (**a**) Butterworth filter of order 3. (**b**) Butterworth filter of order 9. (**c**) Chebyshev type I filter of order 3. (**d**) Chebyshev type I filter of order 9. Solid lines represent theoretical curves, while the band between dotted lines represent the simulations confidence interval for α=0.95. Nyquist frequency $f_{N}=500$ Hz.

**Figure 4 entropy-23-00787-f004:**
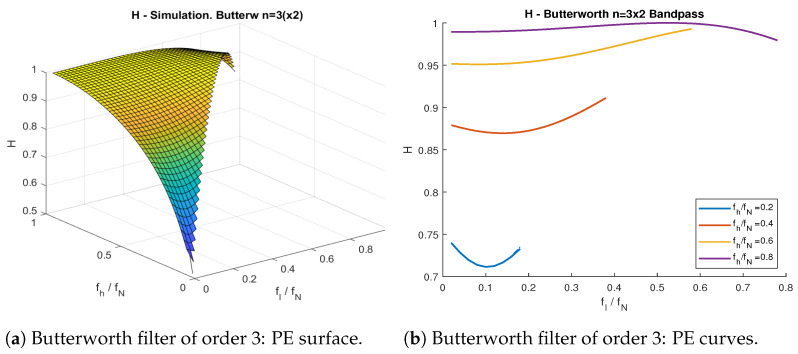
PE as a function of low- $f_{l}$ and high- $f_{h}$ cutoff frequency. (**a**) PE surface representation. (**b**) Curves with respect to $f_{l}$, at set levels of $f_{h}$. The signal is the wGn subject to a Butterworth bandpass filter of order 3.

**Figure 5 entropy-23-00787-f005:**
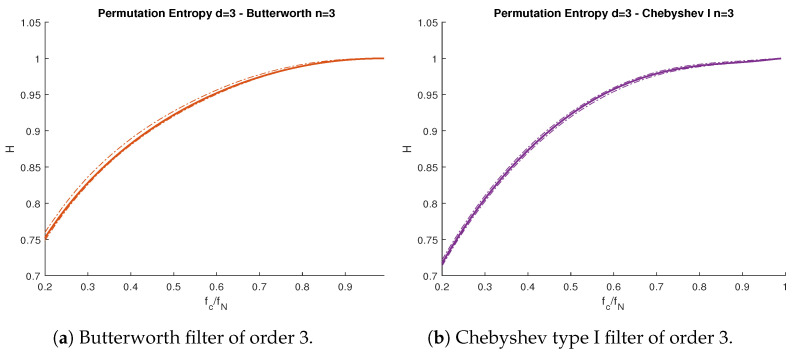
PE vs. normalized cutoff frequency $f_{c}/f_{N}$ for white uniform noise subject to lowpass filters. (**a**) Butterworth filter of order 3. (**b**) Chebyshev type I filter of order 3. Solid lines represent theoretical curves, while the band between dotted lines represent the simulations confidence interval for α=0.95. Nyquist frequency $f_{N}=500$ Hz.

**Figure 6 entropy-23-00787-f006:**
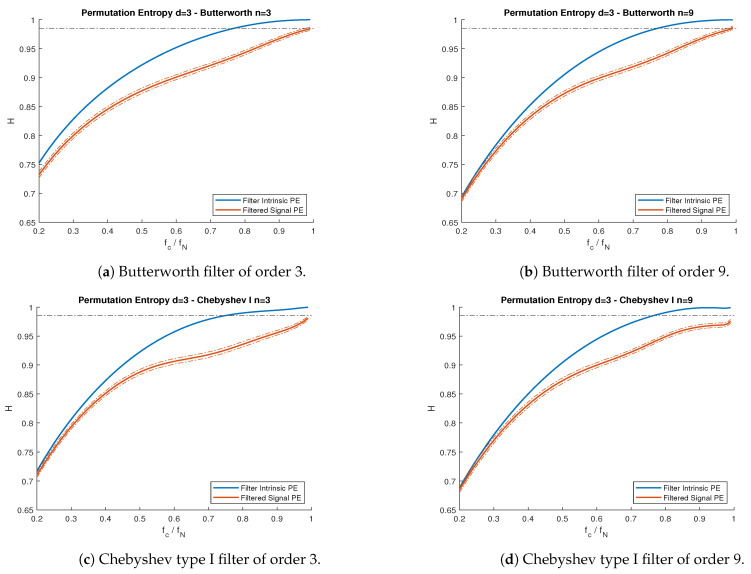
PE vs. normalized cutoff frequency $f_{c}/f_{N}$ for the MA(2) process with $\theta_{1}=0.6$ and $\theta_{2}=0.4$, subject to lowpass filters. (**a**) Butterworth filter of order 3. (**b**) Butterworth filter of order 9. (**c**) Chebyshev type I filter of order 3. (**d**) Chebyshev type I filter of order 9. Blue lines represent the filter’s intrinsic PE, while the red lines are the PE curves for the filtered signals. Solid lines represent theoretical curves, while the band between dotted lines represent the simulations confidence interval for α=0.95. Nyquist frequency $f_{N}=500$ Hz. The black dotted line represents the theoretical PE of the unfiltered MA(2) process.

**Figure 7 entropy-23-00787-f007:**
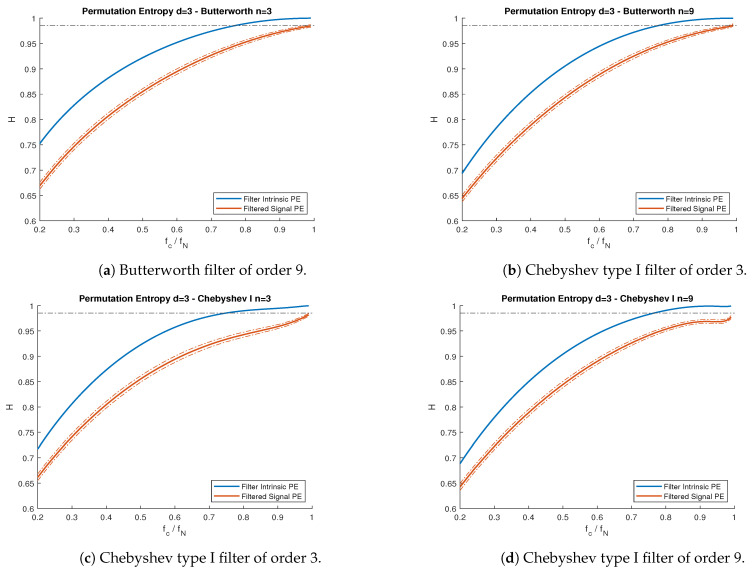
PE vs. normalized cutoff frequency $f_{c}/f_{N}$ for the AR(2) process with $\phi_{1}=0.8$ and $\phi_{2}=0.15$, subject to lowpass filters. (**a**) Butterworth filter of order 3. (**b**) Butterworth filter of order 9. (**c**) Chebyshev type I filter of order 3. (**d**) Chebyshev type I filter of order 9. Blue lines represent the filter’s intrinsic PE, while the red lines are the PE curves for the filtered signals. Solid lines represent theoretical curves, while the band between dotted lines represent the simulations confidence interval for α=0.95. Nyquist frequency $f_{N}=500$ Hz. The black dotted line represents the theoretical PE of the unfiltered AR(2) process.

**Figure 8 entropy-23-00787-f008:**
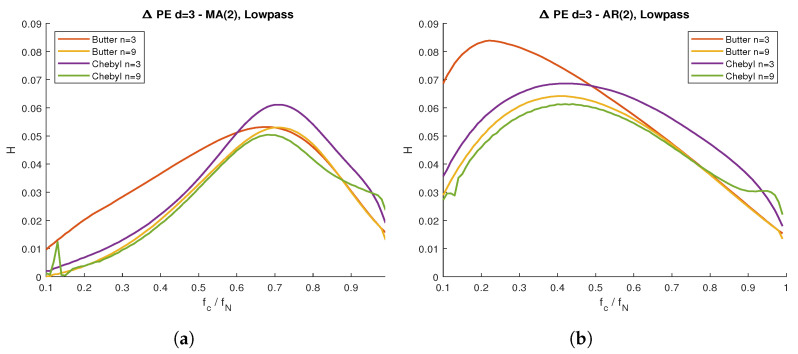
Difference between the filters’ intrinsic PE and filtered ARMA time series, with respect to the normalized cutoff frequency: (**a**) MA(2) series with lowpass filters; (**b**) AR(2) series with lowpass filters. We applied Butterworth (order *n* = 3 and *n* = 9) and Chebyshev type I (*n* = 3 and *n* = 9) filters, for each case.

**Table 1 entropy-23-00787-t001:** Test filter’s parameters.

Type of Filter	Lowpass Parameters	Highpass Parameters	Passband Parameters
Moving average	$L\;\mathrm{such}\;\mathrm{that},\;\frac{\left|\mathrm{sinc}(\pi Lf_{c}/f_{s})\right|}{\left|\mathrm{sinc}(\pi f_{c}/f_{s})\right|}=\frac{1}{\sqrt{2}}$	-	-
Butterworth	n=3	n=3	n=6
Butterworth	n=9	n=9	n=12
Chebyshev type I	n=3, $R_{d}=3$ dB	n=3, $R_{d}=3$ dB	n=6, $R_{d}=3$ dB
Chebyshev type I	n=9, $R_{d}=3$ dB	n=9, $R_{d}=3$ dB	n=12, $R_{d}=3$ dB

L∈N is the length of the moving average filter window; $f_{c}$ is the cutoff frequency, which matches a decrease of 3 dB; $f_{s}$ is the sampling rate; *n* is the order of the respective filter; and $R_{d}$ is the ripple amplitude of the passband in decibels. Subsequently, we applied the PE procedure to each resulting filtered signal, using embedding dimension d=3 and downsampling parameter κ=1.

## Data Availability

Data sharing not applicable—All simulations were generated using MATLAB R2020b.

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
