# Peer review of "The Impact of Linear Filter Preprocessing in the Interpretation of Permutation Entropy"

_entropy, 2021, doi:10.3390/e23070787_

Round 1

Reviewer 1 Report

In this manuscript, the Authors characterize how the permutation entropy (PE) is affected by linear filters. Taking into account that these filters are usually implemented on raw signals as an unavoidable preprocessing step, it is of practical utility to know the potential spurious effects they can have on the PE estimated values. Theoretical results for embedding dimension d=3 are first obtained and, then, numerically simulated tests are included for comparison purposes. To be more precise, several lowpass and highpass linear filters and different types of signals (white Gaussian noise, white uniform noise, a moving average process of order 2, and an autoregressive process of order 2) are considered. The observed agreement between the theoretical and numerical curves is remarkably good confirming and validating the theoretical findings. As the Authors state, it is crucial to characterize the effect of these filters in order to be able to interpret PE obtained values in a more robust and appropriate way. For this main reason, I consider that the manuscript is of particular interest for the audience of Entropy. There are, however, some issues that need to be carefully considered before I can consider that it deserves to be accepted for publication. My comments and suggestions are listed below.

1. The definition of the downsampling parameter tau at lines 77-78 is not clear at all. Please try to clarify. Are you considering different values of tau in the analysis?

2. Related to Fig. 1 and lines 161 & 162: numerical results have been included in Fig. 1 b)?

3. In Section 4.2, where correlated Gaussian signals are analyzed, it seems that fractional Gaussian noise would be a more natural option for studying since they generalize the white Gaussian noise but with temporal correlations. Have you considered this possibility?

4. I am not able to follow the explanation given in lines 213-216. What do you refer to with the "deviations small"?

5. According to the results detailed in Fig. 8, the difference between the PE of the filtered signal and the PE of the filter is a function of the cutoff frequency. Why do you state that this "justifies the use of linear filter preprocessing in order to explore hidden information content" (line 270)?

6. I suggest to include at least one real application for illustrating the usefulness of the obtained results in a practical setting.

Author Response

Thank you very much for your suggestions. We made extensive editing to the manuscript, in order to make it easier to read. More specifically, we addressed your comments as follows:

  1. The definition of the downsampling parameter tau at lines 77-78 is not clear at all. Please try to clarify. Are you considering different values of tau in the analysis?
    1. For the purposes of this work, we left the downsampling parameter tau = 1. This does not affect the theoretical results, since we are able to work with the downsampled series prior to any further filtering. We added this clarification in the manuscript.
  2. Related to Fig. 1 and lines 161 & 162: numerical results have been included in Fig. 1 b)?
    1. We expanded the explanation regarding the model testing using simulations. Indeed, the numerical results are included in figure 1b, but they overlap with the theoretical curve, such that they are not discernible without zooming in. Now figure 1b shows the 95% confidence band for each respective curve, which showcases the results better.
  3. In Section 4.2, where correlated Gaussian signals are analyzed, it seems that fractional Gaussian noise would be a more natural option for studying since they generalize the white Gaussian noise but with temporal correlations. Have you considered this possibility?
    1. Thank you for this suggestion. Indeed fractional Gaussian noise (fGn) provides an interesting extension to analyze correlated signals. We had explored this type of signals in our previous work (Dávalos, A.; Jabloun, M.; Ravier, P.; Buttelli, O. Theoretical Study of Multiscale Permutation Entropy on Finite-Length Fractional Gaussian Noise. 26th European Signal Processing Conference (EUSIPCO) 2018, pp. 1092–1096), albeit only using Multiscaling PE, where the algorithm includes a Moving Average filter as its first step. The differences in theoretical PE are theoretically characterized and discernible. Nonetheless, the differences in PE are small, even for extreme values for the Hurst parameter. In practice, these differences are not easily captured by our simulations. ARMA models, on the other hand, still have the desirable Gaussian properties which allows for the computation of a theoretical curve, which in turn showcases the effect of filtering in figures 6, 7, and 8.
  4. I am not able to follow the explanation given in lines 213-216. What do you refer to with the "deviations small"?
    1. We edited this part in order to express our idea better. Please refer to lines 284-295, where we include the reworked paragraph, along with the old strikethrough text.
  5. According to the results detailed in Fig. 8, the difference between the PE of the filtered signal and the PE of the filter is a function of the cutoff frequency. Why do you state that this "justifies the use of linear filter preprocessing in order to explore hidden information content" (line 270)?
    1. We also edited this part for clarity. You can find both old (strikethrough) and new text in lines 376-390 of the newest version.
  6. I suggest to include at least one real application for illustrating the usefulness of the obtained results in a practical setting.
    1. Thank you very much for the suggestions. We included some references to real signals of Heart Rate Variability (HRV) at the beginning of section 4.2, where ARMA processes are used explicitly to model the HRV signal. Nonetheless, the study of these real series from the perspective of linear filter’s PE would require a full article. For the sake of brevity, we will address these cases in future work.

Please let us know if you have further comments and suggestions.

Best regards,

Antonio Dávalos

Reviewer 2 Report

In this paper, authors analyze the effect of linear filter preprocessing on the Permutation entropy (PE) of the time series for different cases, this work is interesting. However, there are some minor issues that need to be solved before acceptance.

  1. Abstracts should be reorganized, currently a bit confusing.
  2. In line 35, “On the other hand” may not be applied. Is the author trying to make two opposing points?
  3. Understanding the background of this work in the introduction is rather time-consuming, even more so for a casual reader. The author should improve the logic of the introduction, and this work may be further applied to engineering.
  4. The number of dimensions d affects the result of Permutation entropy, how did the authors choose it and on what basis.
  5. Some words in the article are rather colloquial, please revise them.
  6. The conclusions should be better summarized.
  7. Since Equation (10) in Section 3.2 is based on Bandt & Shiha's results, the author should declare that "in Equation (10), the probability of the first pattern is under the condition of embedding dimension d=3 and embedding time delay t=1."
  8. There are some minor issues that need to be modified

In section 2, Permutation entropy should have a time delay factor, please check.

Permutation entropy should have a time delay factor, please check.

For clearance, "corresponding to all the possible ranks." (on line 73) should be "corresponding to all the possible ranks of the segments."

In equation (1), "n<N - (d - 1)t" should be revised as "1≤n≤N - (d - 1)t".

Some recent works about the Permutation entropy would be discussed as follows:

Zhang, H.; Deng, Y. Entropy measure for orderable sets. Information Sciences, 2021, 561: 141-151.

Author Response

Thank you very much for your suggestions. We made extensive editing to the manuscript, in order to make it easier to read. More specifically, we addressed your comments as follows:

  • Abstracts should be reorganized, currently a bit confusing.
    • The abstract has been extensively re-edited.
  • In line 35, “On the other hand” may not be applied. Is the author trying to make two opposing points?
    • This has been corrected as part of the editing.
  • Understanding the background of this work in the introduction is rather time-consuming, even more so for a casual reader. The author should improve the logic of the introduction, and this work may be further applied to engineering.
    • As the abstract, this section was edited accordingly.
  • The number of dimensions d affects the result of Permutation entropy, how did the authors choose it and on what basis.
    • We chose dimension d=3 in order to properly have a PE theoretical benchmark to test our simulations. We explain this particular point in lines 125-128. We added these remarks in sections 3 and 4 for clarity.
  • Some words in the article are rather colloquial, please revise them.
    • We polished the language throughout the document, in order to address this problem.
  • The conclusions should be better summarized.
    • As with the introduction, the conclusions were edited accordingly.
  • Since Equation (10) in Section 3.2 is based on Bandt & Shiha's results, the author should declare that "in Equation (10), the probability of the first pattern is under the condition of embedding dimension d=3 and embedding time delay t=1."
    • Thank you very much for the suggestion. This information is now included in lines 180-182. Furthermore, for the purposes of this work, we use the downsampling parameter tau=1, which we now clarify in lines 108-109.
  • There are some minor issues that need to be modified
    1. “In section 2, Permutation entropy should have a time delay factor, please check.” We now clarify our choice of downsampling parameter, without loss of generality.
    2. “For clearance, "corresponding to all the possible ranks." (on line 73) should be "corresponding to all the possible ranks of the segments."” This remark has now been included
    3. “In equation (1), "n<N - (d - 1)t" should be revised as "1≤n≤N - (d - 1)t"”. This equation has been corrected accordingly.
    4. “Some recent works about the Permutation entropy would be discussed as follows: Zhang, H.; Deng, Y. Entropy measure for orderable sets. Information Sciences, 2021, 561: 141-151.” Thank you for the reference. It was been included.

Please let us know if you have further comments and suggestions.

Best regards,

Antonio Dávalos

Reviewer 3 Report

In general, this is an interesting and timely manuscript. It will attract the attention from the International readership of the Journal. However, a  revision is recommended before a final positive approval.

The idea used by the authors is brilliant. Knowing the properties of the filter, and exploiting the analytical features of the convolution operation, one can explicitly observe the impact of the filter to the PE.

However, the numerical estimate of the PE is not the only important parameter characterising the complexity of a time series. A lot of attention had been focused on forbidden patterns during the last decade. Forbidden patterns also help to assess many important features of the time series.

It would be interesting to explore what impact does linear filter preprocessing    have to forbidden patterns. In other words, the presented analytical investigations could be extended by a number of computational exercises. That would definitely enrich the presentation and the conclusions of this study. 

Author Response

Thank you very much for your comments. We made extensive editing to the manuscript, in order to make it easier to read. More specifically, we addressed your comments as follows:

“It would be interesting to explore what impact does linear filter preprocessing have to forbidden patterns. In other words, the presented analytical investigations could be extended by a number of computational exercises. That would definitely enrich the presentation and the conclusions of this study.”

The analysis of forbidden patterns is indeed an important approach for ordinal analysis, which complements the PE measurements. Based on your comments, we performed some experiments using the logistic map model (Analysis of financial time series through forbidden patterns, Ji 2019). The results using this model have chaotic properties, which are heavily dependent not only the model parameter “r”, but also on the filter’s cutoff frequencies. The following attached figure shows the PE curve of the filtered signal as a function of normalized frequency, for parameters r=1 (blue), r=(1+sqrt(6)) (red), r=(1+sqrt(8)), and r=4 (violet) using a Butterworth lowpass filter of order 3. We show the filter’s intrinsic PE with the black line, for reference.

The proper considerations of the filter effect, in this context, requires extensive theoretical work. This will, indeed, can be properly addressed in further research. We added some comments acknowledging this technique in the manuscript.

Please let us know if you have further comments and suggestions.

Best regards,

Antonio Dávalos

Round 2

Reviewer 1 Report

The Authors have satisfactorily addressed the different issues I have included in the original report. Taking this into account, I consider that the manuscript, in its present form, deserves to be accepted for publication in Entropy.

Reviewer 3 Report

The authors did manage to face all comments and suggestions from the reviewers. Some questions are left for future research, but appropriate discussions in the revised manuscript make it acceptable for publication in the Journal. The recommendation is to accept the revised manuscript.